

# Evaluating Statistical Consistency in the Ocean Model Component of the Community Earth System Model (pyCECT v2.0)

A. H. Baker[1], Y. Hu[2,3], D. M. Hammerling[1], Y. Tseng[1], H. Xu[1], X. Huang[2,3], F. O. Bryan[1], and G. Yang[2,3]

[1]The National Center for Atmospheric Research, Boulder, CO, USA
[2]Center for Earth System Science,Tsinghua University, 100084, China
[3]Joint Center for Global Change Studies, Beijing, 100875, China

*Correspondence to:* Allison H. Baker (abaker@ucar.edu)

**Abstract.** The Parallel Ocean Program (POP), the ocean model component of the Community Earth System Model (CESM), is widely used in climate research. Most current work in CESM-POP focuses on improving the model's efficiency or accuracy, such as improving numerical methods, advancing parameterization, porting to new architectures, or increasing parallelism. Because ocean dynamics are chaotic in nature, achieving bit-for-bit (BFB) identical results in ocean solutions cannot be guar-

anteed for even tiny code modifications, and determining whether model changes are admissible (i.e. statistically consistent with the original results) is non-trivial. In recent work, an ensemble-based statistical approach was shown to work well for statistical consistency testing on atmospheric model data. The general idea of the ensemble-based statistical consistency testing is to use a qualitative measurement of the variability of the ensemble of simulations as a metric with which to compare future simulations and make a determination of statistical distinguishability. Because ocean and atmosphere models have differing

characteristics in term of dynamics and time-scales, we present a new statistical method to evaluate ocean model simulation data that requires the evaluation of ensemble means and deviations in a spatial manner. In particular, the statistical distribution from an ensemble of CESM-POP simulations is used to determine the standard score of any new model solution at each grid point. Then the percentage of points that have scores greater than a specified threshold indicates whether the new model simulation is statistically distinguishable from the ensemble simulations. Both ensemble size and composition are important. Our

experiments indicate that the new POP ensemble consistency test (POP-ECT) tool is capable of distinguishing cases which should be statistically consistent with the ensemble and those which should not, as well as providing a simple, subjective and systematic way to detect errors in CESM-POP due to the hardware or software stack, positively impacting quality assurance for the CESM-POP code.

## 1 Introduction

The Community Earth System Model (CESM) is a popular and fully-coupled climate simulation code (Hurrell et al., 2013) that regularly contributes to the Intergovernmental Panel on Climate Change (IPCC) assessment reports (e.g., Stocker et al., 2013). CESM consists of multiple component models that are coupled together, including component models for the atmosphere, ocean, sea ice, and land. Here, we focus on the Parallel Ocean Program (POP) component of CESM, an extension of the ocean general circulation model originally developed at Los Alamos National Laboratory (Smith et al., 2010). The CESM-POP



solves the three-dimensional (3D) primitive equations for ocean dynamics with hydrostatic and Boussinesq approximations, representing ocean processes across a broad range of spatial and temporal scales. Much new development in CESM-POP is aimed at reducing computational costs (e.g., Hu et al., 2015), but ongoing development of any type in a simulation code requires software quality assurance to ensure that no errors are introduced. The need for some sort of quality assurance to

maintain confidence in the science results is particularly critical for climate models whose simulation output may influence policy decisions with broad societal impact (Carson, 2002; Easterbrook et al., 2011).

Climate models such as CESM are generally large and complex, and the plethora of model configuration options makes them difficult to test exhaustively (Clune and Rood, 2011; Pipitone and Easterbrook, 2012). Further, because of the chaotic nature of climate models, determining whether a difference in simulation results is due to an error or simply to the model's natural vari-

ability can be challenging. Note that a roundoff-level perturbation added to an initial condition or intermediate result can lead to sizable differences in the final result. New developments in CESM-POP, particularly those aimed at improving performance (such as taking advantage of new heterogeneous computing technologies or improving numerical methods), typically result in data output that is not bit-for-bit (BFB) identical to the original code. For CESM-POP, even selecting a different number of cores on the same architecture results in non-BFB identical output. The ability to directly evaluate climate consistency in

the CESM-POP ocean data facilitates the advancement of the code development in general and enables the flexibility to take advantage of new computing (hardware and software) technologies.

The CESM ensemble consistency test (CESM-ECT), recently developed in Baker et al. (2015), addresses the difficulty in comparing climate model outputs via a new ensemble-based tool that evaluates whether a new climate run (e.g., resulting from a hardware or software modification) is statistically distinguishable from an "accepted" ensemble of original (unmodified)

runs. However, the CESM-ECT tool presented in Baker et al. (2015) only evaluates variables from the Community Atmosphere Model (CAM) component of CESM, and the experimental runs do not use a fully-coupled CESM configuration (i.e., rather than POP, the ocean component is the Climatological Data Ocean Model, which contributes sea surface temperature data but does not respond to forcing from the atmosphere component). For clarity, we refer to the general ensemble statistical consistency testing approach for CESM as CESM-ECT. We denote the methodology applicable to the Community Atmospheric Model

component by CAM-ECT, which is a module in the CESM-ECT suite of tools that we are developing. We note that applying the CAM-ECT methodology "as is" directly to ocean data is not feasible because the ocean and atmosphere models greatly differ in terms of their dynamics, spatial scales and time scales. For example, the synoptic scale in the ocean dynamics is one to two orders of magnitude smaller than that in the atmosphere, and the propagation time scale for adjusting the ocean is many orders of magnitude slower than that in the atmosphere, particularly for the deep ocean. Therefore, we have developed a new

approach to provide an ocean-specific methodology for statistical consistency testing, which we denote POP-ECT. Although the new POP-ECT tool is similarly based on using an ensemble of CESM simulations to gauge model variability, it is distinct from CAM-ECT in that the statistical process takes spatial patterns of differing variability into account in the ocean due to a larger time to reach the global quasi-steady state in the ocean than the atmosphere. Further, the smaller number of diagnostic variables available from the ocean model (as compared to the atmosphere) allows for a different approach as well. Finally, note that all experiments in this paper use the publicly available CESM 1.2.2 release.



This paper is organized as follows. The background information is reviewed and discussed in Sect. 2. In Sect. 3, we introduce the new statistical consistency testing methodology for ocean model data, referred to as POP-ECT, as well as the necessary software tools. We evaluate the approach and explore the effect of the simulation length with experimental tests in Sect. 4.

Finally, we explore the new approach's sensitivity to ensemble size in Sect. 5 and provide concluding remarks in Sect. 6.

## 2  Background discussion

### 2.1  Current ocean model quality assurance testing

The current POP-specific quality assurance test, referred to as POP-RMSE, is a simple test that evaluates whether the CESM-POP code was successfully ported to a new architecture and aims to discover issues related to a new machine's hardware or

software stack. This test consists of running five days of a specified case on a new machine and then comparing the output to that of a standard dataset released by the National Center for Atmospheric Research (NCAR). The comparison is done only for the sea surface height (SSH) field via a root-mean-square error (RMSE) calculation, which measures the difference between the two datasets, $X_0$ and $X_1$, each containing $n$ grid point values:

$$RMSE(X_0, X_1) = \sqrt{\frac{1}{n} \sum_{i=1}^{n} (X_1(i) - X_0(i))^2}.$$

The rate of growth of RMSE was compared to the growth between two reference cases in which the convergence criteria for the solver was changed by one order of magnitude.

The simple methodology in POP-RMSE is convenient for evaluating CESM output on a new machine, but it is far less comprehensive than CAM-ECT for atmospheric simulation data. For example, POP-RMSE was unable to quantify the small climate state changes due to recent linear solver modifications in CESM-POP in Hu et al. (2015). In Hu et al. (2015), the

authors replaced the default preconditioned conjugate gradient (PCG) solver for the barotropic mode of POP by an alternative preconditioned Chebyshev-type iterative (P-CSI) solver to enhance the computational performance. While the P-CSI solver had considerably lower communication costs for high-resolution simulations than PCG, showing that the use of an alternative solver did not negatively impact the ocean simulation results was critical for acceptance. Therefore, in Hu et al. (2015), to gauge the effectiveness of the POP-RMSE test in detecting solver differences over time, monthly data was collected for 36 months

from a CESM-POP $1°$ resolution case with multiple convergence tolerances between $10^{-10}$ and $10^{-16}$. Figure 1 displays the RMSE between the strictest case ($10^{-16}$) and the other tolerances listed in the figure's legend for the temperature field. Despite the range in convergence tolerances used, Fig. 1 gives scant evidence of any solver-induced error (Hu et al., 2015).

### 2.2  Ensemble consistency testing

Because the RMSE approach did not elucidate differences between convergence tolerances, Hu et al. (2015) adapted the

ensemble approach for POP data that was initially described in the context of data compression in Baker et al. (2014) (a precursor to the CESM-ECT approach). Note that the ensemble in Baker et al. (2014) was created by perturbing an initial





condition. However, changing the convergence tolerance for the linear solver or introducing an alternative solver, as in Hu et al. (2015), can be categorized as perturbing a forcing term. Of interest is whether the broad effect of perturbing an initial condition captures the effect of perturbing a forcing term at each time step. As in Caya et al. (1998), we represent POP's

time-stepping progress by the following simple equation with minimal components:

$$\frac{\partial X(t)}{\partial t} = \alpha X(t) + G, \tag{1}$$

where $G$ is (constant) forcing term, $\alpha$ is a constant, $X(t)$ is a dependent variable that is a function of time, and $X(0) = X_0$. The corresponding analytical solution to Eq. (1) is

$$X(t) = X_0 e^{\alpha t} - \frac{G}{\alpha}. \tag{2}$$

For this exercise, we assume that $\alpha$ is a purely imaginary number, so that $\alpha X(t)$ is an oscillation (we need not address a damping mechanism).

First consider a perturbation to the initial condition:

$$X_0' = X_0 + P,$$

where $P$ is a constant. Then the analytical solution to Eq. (1) with the perturbed initial condition is

$$X_P(t) = (X_0 + P)e^{\alpha t} - \frac{G}{\alpha}, \tag{3}$$

or equivalently,

$$X_P(t) - X(t) = Pe^{\alpha t}. \tag{4}$$

Equation (4) indicates that the error between the original solution and that with the perturbed initial condition is oscillatory. Now consider a perturbation to the forcing term:

$$G' = G + F,$$

where $F$ is a constant. Then the analytical solution to Eq. (1) with the perturbed forcing term is

$$X_F(t) = X_0 e^{\alpha} - \frac{G + F}{\alpha}, \tag{5}$$

which reduces to

$$X_F(t) - X(t) = \frac{F}{\alpha}. \tag{6}$$

Equation (6) indicates that the error due to perturbing the forcing term is a constant. Therefore, we can use the magnitude of the error caused by perturbing an initial condition in Eq. (4) to measure the error caused by perturbing the forcing term in Eq. (6). For example, Fig. 3 illustrates a case where the initial perturbation and forcing perturbation have similar effects on the



final result in Eq. (2), and the magnitude of error induced by a perturbation to the forcing term is less than that of perturbing the initial condition. The top plot shows the analytic solutions generated from Eq. (3) and Eq. (5), and the bottom plot shows the errors from Eq. (4) and Eq. (6) corresponding to the values of $P$ and $F$ given in the figure's legend. Therefore, the choice

in Hu et al. (2015) to evaluate solver-induced variability based on an ensemble created via a perturbation in the initial ocean temperature field is reasonable.

In particular, Hu et al. (2015) created an ensemble consisting of 40 runs of 36 months in length that differed by an $\mathcal{O}(10^{-14})$ perturbation in the initial ocean temperature field. Next the root-mean-squared Z-score between the new case, $\tilde{X}$, and the ensemble of test cases was calculated as follows. If size of ensemble $E$ is denoted by $N_{ens}$, then at each grid point $i$, $N_{ens}$

values exist for each variable $X$. The average and standard deviation at each grid point $i$ for the $N_{ens}$ ensemble members of $E$ are denoted by $\mu_i$ and $\sigma_i$, respectively. Recalling that $n$ is the number of grid points, for each variable $\tilde{X}$ in the new run, the root-mean-square Z-score of $\tilde{X}$ as compared to ensemble $E$ is

$$RMSZ(\tilde{X}, E) = \sqrt{\frac{1}{n} \sum_{i=1}^{n} (\frac{\tilde{x}_i - \mu_i}{\sigma_i})^2}. \tag{7}$$

Figure 2 shows the results of the RMSZ scores for the same five selected convergence tolerances as in Fig. 1, and now the error

induced by the least strict tolerances is more evident. Based on this result in Fig. 2, Hu et al. (2015) evaluated the suitability of a new solver in CESM-POP.

## 2.3  Ensemble consistency testing for the Community Atmosphere Model

The CESM-ECT approach presented in Baker et al. (2015) uses an ensemble to quantify the natural variability of the CESM model's climate and then compares new simulations (resulting from a software, hardware, or non-BFB change) against the

ensemble distribution. The key idea is that if the output data from a new simulation is not statistically distinguishable from the ensemble of runs, then the new run is deemed "consistent". Statistical consistency is a key ingredient in the quality assurance aspect of model code verification (Oberkamf and Roy, 2010). The CESM-ECT approach was first applied to history output from the Community Atmosphere Model (CAM) component, resulting in the CAM-ECT tool. CAM data were a logical first target as the time scales for changes to propagate throughout the atmosphere are shorter compared to other CESM components

and the CAM contains a large number of independent variables that cover the whole globe.

As described in Baker et al. (2015), the ensemble for CAM-ECT is created on a trusted machine with an accepted version and configuration of CESM. The ensemble consists of 151 simulations of 1-year in length that differ only in a random perturbation of the initial atmospheric temperature field of $\mathcal{O}(10^{-14})$. The output data contains only the annual averages at each grid point for the number of specified variables from CAM, denoted by $N_{var}$, which represent whole atmospheric fields (by default,

$N_{var} = 120$). The CAM-ECT tool creates a statistical distribution that characterizes the ensemble using principal component analysis (PCA) on the global area-weighted mean distributions for each variable across the ensemble. The distribution of principal component (PC) scores are retained for comparison with new simulation runs. A small set of new runs (generally



3) either passes or fails based on the number of PC scores that fall outside a specified confidence interval (typically 95%). Parameters specifying the pass/fail criteria can be tuned to achieve a desired false positive rate for the test.

## 2.4 Motivation

While the solver verification work in Hu et al. (2015) was illuminating and adequate for the task at hand, it prompted several questions that motivated our work to develop a more comprehensive technique such as CAM-ECT for evaluating CESM-POP data. First, a key consideration for a more general POP verification tool was the concern over spatial variability in the ocean, which is much more pronounced than in the atmosphere. Both CAM-ECT as well as the RMSZ strategy in Hu et al. (2015) evaluate the differences in terms of spatial averages, and it was unclear whether this approach would be sufficient to detect any accuracy issue or model error in the ocean. Second, because CESM-POP has fewer independent diagnostic variables than the CAM, we re-visit the question of appropriate metrics for consistency evaluation. Third, the selection of different metrics prompted further examination of the appropriate ensemble size. While an ensemble of size 40 ensemble was sufficient for detecting linear solver errors in Hu et al. (2015), this dimension was not thoroughly explored in a more general context, particularly in light of the recommended ensemble size of 151 for CAM-ECT. Finally, the difference in temporal scales between the ocean and atmosphere prompted us to investigate the required length of ensemble runs. Because the time-scales in the ocean are slower than in the atmosphere, intuitively a longer ensemble length is needed for statistical consistency testing with CESM-POP. Note, though, that the initial results in Fig. 2 suggest that this presumption may not be true. The RMSZs continue to decrease with time after a few months when the tolerances are small enough (i.e., tolerance is smaller than $10^{-12}$), suggesting the required length of simulation merits further investigation. In fact, Fig. 2 shows that this particular 1-degree ocean data is relatively deterministic and the initial differences begin to damp out (grow) after the first year when the tolerances are small (unrealistically large).

## 3 A new statistical consistency test for POP

Expanding the CESM-ECT suite to include a consistency test for CESM-POP, which we denote POP-ECT, data requires determining an appropriate ensemble to represent ocean model variability and developing a methodology to address ocean-specific characteristics. We first discuss the ensemble creation. We then describe the new testing procedure.

### 3.1 POP ensemble construction

We evaluate whether differences in CESM-POP output data are statistically significant (i.e., indicative of a changed climate state) by comparing to an ensemble of simulations representing an accepted ocean state. Therefore, the first stage in applying the CESM-ECT approach to CESM-POP data is creating an appropriate ensemble. Clearly, the ensemble composition is critical to an effective test, and simulations should be produced on an accepted machine with an accepted version of CESM. Note that as compared to CAM, the ocean model has fewer independent diagnostic variables: temperature (TEMP), sea surface height





(SSH), salinity (SALT), and the zonal and meridional velocities with respect to the model grid (UVEL and VVEL, respectively). Of these five variables, SSH is 2D and the remainder are 3D.

For POP-ECT, we create an ensemble $E$ of $N_{ens}$ simulations, denoted by $E = \{E_1, E_2, \ldots, E_{N_{ens}}\}$, that differ only by an

$\mathcal{O}(10^{-14})$ perturbation to the initial ocean temperature field. This initial perturbation size is on the order of double-precision round-off error and should not be climate-changing. The size of the ensemble must be sufficient to create a representative distribution, but as small as possible to avoid high computational cost. While we address in detail our choice of ensemble size later in Sect. 5, we use $N_{ens} = 40$ for the discussion in this section and the experiments in the next. Our experiments indicate that this choice of ensemble size adequately represents the natural variability in the ocean for testing purposes. The ensemble

simulation data consists of monthly temporal averages at each grid point $i$ for the 5 POP diagnostic variables: TEMP, SSH, SALT, UVEL, and VVEL. Each of these variable datasets $X$ contains $N_X$ grid points and is denoted by $X = \{x_1, x_2, \ldots, x_{N_X}\}$, where $x_i$ is a scalar monthly average at grid point $i$. Data is collected for $T$ months (i.e., $T$ time slices of monthly data).

The next step in CESM-ECT approach is to characterize the ensemble distribution in a qualitative way to facilitate the evaluation of new runs. This statistical description of the ensemble is stored in a so-called ensemble summary file and is

associated with the CESM softare tag used to generate the ensemble simulations. The history files from the $N_{ens}$ simulations do not need to be retained once the summary file has been created. Recall that for CAM data, CAM-ECT calculates the global weighted-area mean for each variable from the annual temporal averages available at each grid point. This calculation results in a distribution of $N_{ens}$ global means for each variable. However, this approach of using global weighted-area means will not be appropriate for ocean model data, as ocean variability is less uniform across the grid than atmospheric data.

For example, consider an ensemble of $N_{ens} = 40$ CESM simulations with a $1°$ resolution CESM-POP grid run for $T = 36$ months. In Fig. 4, we show spatial plots of the standard deviations of the sea surface temperature (SST), across the ensemble members after 1, 12, 24, and 36 months. Note that the SST is simply the top layer of the 3D variable TEMP (more precisely, the top 10 meters of upper ocean). Figure 4 shows that the standard deviation is far from uniform with orders of magnitude differences across the grid (see the color bar scale). Also notable is the change from 1 month to 12 months associated with

the model instability associated with the development of hydrodynamic instability of the flow as the model spins up. At the end of one year, the larger standard deviation in the tropical regions suggest a larger uncertainty due to the growth of tropical instability waves (Legeckis, 1977) in the ensembles. The large variability can be easily enhanced in the equatorial regions because of the finer resolution in the tropics (approximately $1/3°$). There are also some pools of large standard deviation in the downstream of the major ocean current systems. The change from 12 months to 36 is more subtle, indicating the associated

physical instability may not grow further due to the ocean dissipation. Given the range in variability evident in Fig. 4, the RMSZ score strategy as used for verification in Hu et al. (2015) (and discussed in Sect. 2) may include the uncertainty introduced by the actual physical instability in regions with large variability (e.g., the equatorial Pacific) and may unnecessarily flag potential errors in regions with little to no variability due to the denominator in Eq. (7). Note the range of variability shown in Fig. 4 is certainly resolution dependent, and the results in Fig. 4 are specific to the rather dissipative low-resolution ocean model (e.g., $1°$ CESM-POP) used in most climate studies.




Therefore to create an ensemble statistical consistency test that is robust for CESM-POP data, we must create a distribution describing the ensemble that contains spatial as well as temporal information. In particular, POP-ECT creates an ensemble file with $T$ monthly time slices of CESM-POP data that contains:

– $N_{var} \times N_X \times T$ monthly mean values across the ensemble at each grid point $i$ ($\mu_i$)

   – $N_{var} \times N_X \times T$ standard deviations of ensemble monthly mean values at each grid point $i$ ($\sigma_i$),

which is to say that we retain the ensemble mean and standard deviation at each of $N_x$ grid points (note that $N_X$ depends on whether $X$ is a 2D or 3D variable) for the specified number of months, e.g. $T = 36$.

Finally, we note that without special treatments, the CESM-POP could generate unrealistic salinity distributions in the closed

marginal seas because there is no appreciable freshwater feedback between the freshwater and salinity (e.g. Hudson Bay, the Mediterranean Sea). The current CESM-POP imposes a strong freshwater restoring in the uncoupled simulation and a marginal sea freshwater balancing strategy in the coupled simulation. These specific treatments can maintain a salinity balance but act as artificial forcings to the model dynamics. Therefore, in this work we do not address the complexity of how to do verification properly in the marginal seas and instead restrict our attention to the open oceans.

## 3.2  Testing procedure

Given the POP-ECT summary file, we determine whether new simulation output data (e.g. from a code modification, a new machine, a new compiler option) is statistically consistent with the ocean climate categorized by the reference ensemble distribution as follows. Because we only have five diagnostic variables that are well-understood, we do not need to use PCA. Instead, we take the approach of evaluating the standardized difference between the ensemble and the new run *at each grid*

*point*. For each grid point $i$ and each new variable $\tilde{X}$, we calculate the distance between $\tilde{X}$ and the ensemble data via a standard Z-score measurement for a given monthly time slice $t$. In particular, given the values of $\tilde{X}$ at time $t$, $\tilde{X} = \{\tilde{x}_{1,t}, \tilde{x}_{2,t}, \dots, \tilde{x}_{N_X,t}\}$, the Z-score at grid point $i$ for variable $\tilde{X}$ at time $t$ is

$$Z_{\tilde{x}_{i,t}} = \frac{\tilde{x}_{i,t} - \mu_{i,t}}{\sigma_{i,t}},$$

where $\mu_{i,t}$ and $\sigma_{i,t}$ are the ensemble mean and standard deviation respectively, at grid point $i$ for variable $X$ at the specified

month $t$ as specified in the ensemble summary file.

Now for a particular time slice $t$, we drop all subscripts $t$ from relevant variables, e.g. the Z-score becomes $Z_{\tilde{x}_i}$. We define an allowable tolerance $tol_Z$ for the Z-score at each point, meaning that if $Z_{\tilde{x}_i} > tol_Z$, then point $i$ is denoted a "failed" point. Recall that a Z-score indicates the number of standard deviations away from the mean, and a large Z-score indicates that the new case is far from its climate state in the ensemble. Next we look at the overall percentage of grid points that have passing

Z-scores, defining the Z-score Passing Rate (ZPR) for variable $\tilde{X}$ as:

$$ZPR_{\tilde{X}} = \frac{\#\{i \mid \tilde{x}_i \in \tilde{X} \, \wedge \, |Z_{\tilde{x}_i}| \, \leq \, tol_Z\}}{\#\{i \mid \tilde{x}_i \in \tilde{X}\}}. \tag{8}$$





To make an overall determination of whether variable $\tilde{X}$ passed, we set a minimum threshold for the ZPR ($min_{ZPR}$). In particular, if $ZPR_{\tilde{X}} \geq min_{ZPR}$ then variable $\tilde{X}$ passes. By default, the Z-score tolerance is $tol_Z = 3.0$, and the ZPR threshold is $min_{ZPR} = 0.9$. In other words, 90% of the new values for variable $\tilde{X}$ must be within 3.0 standard deviations of the ensemble mean ($\mu_i$) at each grid point $i$ for $\tilde{X}$ to "pass". This process is repeated for all five independent diagnostic variables, and all variables must pass for the overall simulation to be deemed statically consistent.

Note that the calculated Z-scores change with simulation length. Because of the longer time-scales present in the ocean, we ran the CESM simulations for most of the experiments in the paper for 36-months. In addition, we output monthly time slice data for POP-ECT (as opposed to the annual temporal mean for CAM-ECT) to determine whether the ensemble ocean states stabilize (or not) over time.

As will be evident in the following section, the ZPRs generally become stable after a few months, and the stability trends across the diagnostic variables are similar. Therefore, in addition to picking a suitable Z-score tolerance and passing rate, we choose a checkpoint ($t_C$) at which to evaluate the new run result (instead of checking at all $T$ months of data). Note that the length of the ensemble simulations does not need to be longer than $t_C$.

## 3.3 Software tools

To make our new POP-specific testing methodology accessible to both users and developers, we added POP-ECT to the existing CESM-ECT suite of Python tools (pyCECT v2.0) , which are included in the CESM public releases. The CESM-ECT Python tools include the tools that create the CESM module-specific ensemble summary files as well as pyCECT, which performs the statistical consistency test using the specified ensemble summary file. Because the POP-ECT summary file is distinct from the CAM-ECT summary file, we created the parallel Python code pyEnsSumPop to generate the POP-ECT summary files. In particular, from an ensemble of CESM-POP simulation output files, pyEnsSumPop creates the ensemble summary file (in parallel) containing the ocean model statistics as described in Sect. 3.1. The CESM Software Engineering Group creates a new ensemble of POP simulation data as needed, which currently coincides with the release of a software tag that contains modifications known to alter the climate from the previously tagged version's climate. The appropriate POP-ECT ensemble summary files are included in development and release tags for CESM as noted in Sect. 7. Given a POP-ECT summary file, a user or developer can then evaluate "new" simulation data for consistency using the pyCECT Python tool, which is now able to evaluate results based on either the POP-ECT or CAM-ECT methodology. New CESM-POP simulation data to be evaluated may be the result of using a new architecture or a different compiler option, making a code modification, or changing the input data deck. pyCECT evaluates whether the new ocean model simulation results are statistically consistent with the specified POP-ECT ensemble and issues an overall "pass" or "fail" designation. In addition, the Z-score passing rate is given for each ocean model variable at the selected checkpoint time $t_C$.



## 4  Experiments

The primary objective of this section is to evaluate the new POP-ECT tool on CESM-POP simulation data with a series
of experiments on configurations with expected outcomes, including revisiting the effect of changing the barotropic solver
convergence tolerance. Experiments were run with the CESM 1.2.2 release, using CESM-POP for the active ocean component,
the CICE model for the active sea ice component, and data-driven atmosphere and land components. In addition, we use the
climatologically-averaged atmospheric forcing (one-year repeating forcing) framework for ocean-ice simulations. Therefore,
there are no year-to-year corresponding events (such as El Niño Southern Oscillation), and the variance in the equatorial Pacific
may be artificially suppressed. (Note that this particular CESM component configuration is referred to in CESM documentation
as a "G_NORMAL_YEAR" component set). The CESM grid resolution was "T62_g16", which corresponds to a $1°$ grid ($320 \times$
384) for the ocean and ice components, with 60 vertical levels and a displaced Greenland pole. Simulations were run on 96
processor cores (unless otherwise specified) on the Yellowstone machine at NCAR.

For these experiments, we evaluate 36 months of data as opposed to a single time slice to provide insight as to how the ZPRs
vary over time and guide the selection of $t_C$. Further, to illuminate the relationship between the Z-score and simulation month
in terms of ZPR and guide the selection of $tol_Z$ and $min_{ZPR}$, we utilize a Response Surface Methodology (RSM) (e.g., Box
and Draper, 2007). That is, we provide plots of the response surfaces for variable $\tilde{X}$ where the percentage of grid points $i$ that
meet the Z-score tolerance criteria, $Z_{\tilde{x}_i} > tol_Z$, are shown with a cumulative distribution function (CDF) for a range of $tol_Z$
values and simulation months. Finally, as noted previously, we find an ensemble size of 40 to be sufficient for our experiments,
but we further explore and discuss the ensemble size parameter selection in Sect. 5.
For simplicity, we show results for temperature (TEMP) and sea surface height (SSH). Though we analyzed the other
variables as well, these two are representative of the ocean system model in general as SSH is related to ocean circulation
dynamics and TEMP is determined by model scalar transport.

### 4.1  Barotropic solver convergence tolerance

First we use the newly enhanced CESM-ECT to revisit the effect of changing the barotropic solver convergence tolerance,
as discussed in Sect. 2 in reference to the work in Hu et al. (2015). The default barotropic solver convergence tolerance in
CESM-POP is $10^{-13}$, and we ran experiments with convergence tolerances ranging from $10^{-9}$ to $10^{-16}$, outputting monthly
temporal averages at each grid point for 36 months. We expect convergence tolerances tighter than the default $10^{-13}$ to result
in a consistant climate, but looser tolerances to introduce some error.

Response surfaces for TEMP and SSH are given in Fig. 5 and Fig. 6, respectively. Each figure contains four response
surfaces: the original default convergence tolerance ($10^{-13}$) and a tighter tolerance ($10^{-16}$) in the top two subplots and looser
tolerances ($10^{-10}$ and $5.0 * 10^{-9}$) in the bottom two subplots. For each response surface, the x-axis indicates the simulation
month (ranging from 1 to 36), and the y-axis indicates the range of Z-score values used for $tol_Z$ when calculating the percentage
of grid points that fall below the Z-score tolerance, i.e. ZPR in Eq. (8) . The color bar indicates the ZPR as a percentage in
increments of 10%. The response surface plots are useful for evaluating various combinations of options for $tol_Z$ and $min_{ZPR}$.





For example, consider the effect on variable TEMP of modifying the solver convergence tolerance. The upper left subplot in Fig. 5 indicates that for the original convergence tolerance ($10^{-13}$), 90% of all grid points had a Z-score of less than 2.0 at all simulation months. In contrast, the subplot below for $10^{-10}$ shows that after the first 9 simulation months, 90% of the grid points have a Z-score less 3.0, and by 12 months, between 70 and 80 percent of the grid points have Z-scores less than 2.0. Further loosening the convergence tolerance to $5.0 * 10^{-9}$ as in the lower right subplot shows pronounced errors in terms of the relatively low ZPR percentages. If we turn our attention to SSH in Fig. 6 for the same four convergence tolerances, the overall trends are similar. In particular, for $10^{-13}$, 90% of all grid points have a Z-score of less than 2.0 at all simulation months (except month 6). Similarly to TEMP, the subplot for $10^{-10}$ shows that errors have been introduced and errors are even more pronounced for $5.0 * 10^{-9}$. A notable difference between the response surfaces for TEMP and SSH is that the plots for temperature are smoother over time because diffusion is an important process in the temperature calculation.

If we fix the Z-score tolerance for the data shown in Fig. 5 and Fig. 6, we can more easily evaluate the ZPR. Consider setting $tol_Z = 3.0$, a rather conservative choice. Fig. 7 illustrates the percentage of grid points with Z-scores that exceed $tol_Z = 3.0$ ( i.e. *fail*) for both TEMP and SSH. If we choose a ZPR threshold of $min_{ZPR} = 0.9$, which corresponds to a 10% failure rate in Fig. 7, it is clear that a convergence tolerance of $10^{-10}$ is borderline in terms of passing or failing (and therefore should not be used in practice). Whereas tolerances tighter than $10^{-10}$ have low failure rates and appear statically consistent with the original tolerance for both variables. This plot in Fig. 7 is of interest as well as it nicely demonstrates that as the convergence tolerance becomes less strict, the number of grid points exceeding the Z-score tolerance increases. This result is much clearer than in Hu et al. (2015).

## 4.2 Processor layouts

While CESM simulations that are identical except for differing numbers of CESM-POP processor cores yield non-BFB identical results, the results from such simulations should represent the same climate state (i.e., they should not be statistically distinguishable). Here we verify that such simulations definitively pass CESM-ECT. Recall that the simulations comprising our CESM-ECT ensemble were run on 96 cores. We ran additional simulations on 48, 192, and 384 cores. Note that we are not using threading in CESM-POP at this time.

The response surface plots for 96 cores (labeled "original") and 384 cores are the top subplots in Fig. 9 and Fig. 10 for TEMP and SSH, respectively. These plots show that for both core counts, 90% of all grid points have a Z-score of less than 2.0 for nearly all simulation months, and as expected, there is little discernable difference between the two core counts for both variables. As before, we fix the Z-score tolerance at $tol_Z = 3.0$ and show the Z-score failure rates for TEMP with all four core count options (48, 96, 192, and 384) in Fig. 8. As anticipated, the failure percentages are quite low (below 1.2%) for all configurations at all monthly time slices, confirming that differences in simulation output due to varying the core count in CESM-POP are not statistically significant and correctly identified as such by the new CESM-ECT methodology. Note that the corresponding plot for SSH is not provided as is looks similarly good in terms of very low failure rates.



### 4.3 Physical parameters

Now we change two physical parameters expected to alter the ocean climate from the tracer equations: the tracer's vertical mixing coefficient for convective instability and the tracer advection scheme. Results from these modifications should fail the
CESM-ECT. First, by default, the vertical mixing coefficient for convective instability (*convect_diff*) is set to be *convect_diff* = 10,000 for the tracer mixing coefficient in the 1°CESM-POP configuration. We increase this parameter by factors of 2, 5, and 10, which is expected to increase the vertical mixing in the ocean interior when the density profile is unstable. This should noticeably impact the CEM-POP results due to the different mixing property. Second, we change the POP tracer advection scheme (*t_advect_ctype*) from the default 3rd-order upwind scheme (*upwind3*) to the Lax-Wendroff scheme with 1D flux
limiters (*lw_lim*). This change is also significant and should lead to a different climate state because the associated diffusion and dispersion errors differ.

The response surface plots for increasing *convect_diff* by a factor of 10 are given in the lower left subplots in Fig. 9 and Fig. 10 for TEMP and SSH, respectively. This change clearly affects the climate state significantly, particularly as compared to changing the CESM-POP core count to 384 as depicted in the upper right subplot in both figures. In fact, the impact on TEMP
of increasing *convect_diff* in Fig. 9 is almost as strong as changing the solver convergence tolerance to $10^{-9}$ in Fig. 5. The change of the advection scheme also leads to different climate state, evident in the lower right subplots in Fig. 9 and Fig. 10 for TEMP and SSH, respectively. Note that the Z-scores at nearly every grid point are failing.

The Z-score failure rates for $tol_Z = 3.0$ are shown in Fig. 11 for advection scheme change as well as all the modifications to the tracer vertical mixing coefficient for convective instability. If we choose a ZPR threshold of $min_{ZPR} = 0.9$, which
corresponds to a maximum of 10% failure rate, then doubling the vertical mixing coefficient (*convect_diff*\*2) is borderline in terms of passing or failing. The remaining tests clearly fail for both TEMP and SSH, as expected. Based on our experiments thus far, choosing a Z-score tolerance of $tol_Z = 3.0$ and a ZPR threshold of $min_{ZPR} = 0.9$ yields the expected outcome, and these parameter settings are the default for the pyCECT tool.

### 4.4 Simulation length

Our experiments in Fig. 11 indicate that the percentage of grid points with failing Z-scores differs little from month to month after the first 12 months for both TEMP and SSH. This conclusion can also be reached from Fig. 9 and Fig. 10 for TEMP and SSH, respectively. In particular, the response of SSH to the initial temperature perturbation is largely stabilized after 12 months. The SSH may be affected through the circulation change resulting from the change of density stratification. Based on our experimental results, evaluating the output at a single well-chosen checkpoint time $t_C$ appears reasonable.
We generally choose $t_C = 12$ to minimize the computational requirements of creating the ensemble for each candidate CESM tag. (The ensemble simulations runs can be length $t_C$.) Consider the surface plots for month $t_C = 12$ in Fig. 12 that illustrate the Z-score values for SST as compared to the ensemble for four different model configurations. The top subplot is the original case. The second plot from the top is the result of changing the number of CESM-POP processor cores to 384, which resembles the topmost plot as expected. While the patterns are not identical for the upper two plots, the Z-score




magnitudes and distributions are similar, indicating a degree of statistical consistency when changing the number of processors. In contrast, increasing the tracer mixing coefficient for convective instability by a factor of 10 was shown to change the climate state in Sect. 4.3, and this result is clearly evident in the Z-score at month 12 in the third plot from the top of Fig. 12. Finally,

the bottom subplot in Fig. 12 indicates a largely altered climate state due to the use of a different advection scheme, which corroborates the substantial effects seen in Fig. 9 and Fig. 10. Using a different advection scheme significantly changes the numerical dissipation and diffusion associated with the scheme (Tseng, 2008) and effectively influences the circulation pattern and structure in the ocean model (e.g., Tseng and Dietrich, 2006). In particular, the Lax-Wendroff scheme with flux limiters can introduce excessive numerical mixing which may interact with the physical mixing of temperature and salinity, though it

can result in a much smoother solution in general.

## 5  Ensemble size

The size (i.e., number of members) of the ensemble must be large enough to sufficiently capture ocean model variability, but as small as possible for computational efficiency. In this section, we discuss the sensitivity of POP-ECT to ensemble size. We setup experiments to determine the false positive rate associated with multiple ensembles sizes as follows. First, we generate

a total of 80 ensemble members that differ by an $\mathcal{O}(10^{-14})$ perturbation to the initial ocean temperature field. Second, from the 80 members, we remove 10 to use as our test set. Next, from the remaining 70 members, we create ensembles of sizes 10, 20, 30, 40, 50, and 60. In particular, for each ensemble size, we do 100 random draws for each ensemble size from the set of 70 members, resulting in 100 distinct ensembles corresponding to each ensemble size. Then for each ensemble size, we run POP-ECT at $t_c = 12$ months for each of the 10 members of the test set with all 100 ensembles of that size, resulting in

1000 tests per ensemble size. We consider the measured experimental failure rate to be the type I error, or "false positive" rate. Because the test set and the ensemble members are all drawn from the larger 80 member collection that represents a statistically consistent climate, the Z-score failure rate would ideally be as low as possible.

Figure 13 shows the results of performing these experiments for variables TEMP and SSH. The x-axis indicates the ensemble size, and the y-axis indicates the Z-score failure rate. For each ensemble size, the squares denote the mean and the error bars

indicate one standard deviation of uncertainty. As expected, as the ensemble size increases, the false positive rate decreases and the range of uncertainty shrinks. However, increasing the ensemble size has diminishing returns; the improvement in false positive rate when using 20 instead of 10 members is much greater than the improvement gained by using 60 instead of 50 members. We choose an ensemble size of 40 as improvement beyond that is marginal and we balance a low false positive rate with keeping the cost of ensemble generation low.

## 30  6  Conclusions and Future Work

Because the CESM-POP ocean model is widely-used and critical to many climate simulations, assuring its quality is critical. However, the chaotic nature of ocean dynamics often leads to simulation results that are not identical in the presence of minor





differences, such as a change in processor core count for the simulation. Therefore, the ability to easily determine whether differences in model results are statistically significant is important to both climate scientists and model developers. The ensemble methodology developed for evaluating consistency with atmospheric data, CAM-ECT, was not appropriate for ocean

simulation data based on its differing characteristics. Therefore, we developed a new ocean model-specific methodology for statistical consistency testing, POP-ECT, that allows for the subjective detection of statistically significant changes in CESM-POP. Together with the new methodology, using an appropriately-sized ensemble is critical as well. Our experiments indicate the appropriateness of the new approach for detecting differences in the model ocean state. The addition of POP-ECT to the CESM-ECT suite of tools has greatly enhanced the capability to ensure quality CESM simulations.

We plan to extend this work in a number of ways. First, the existing spatial approach lends itself to the examination of regional ocean diagnostics. For example, named oceans could be identified individually as the source of failure if the global test fails. Enabling the move from coarse- to fine-grain diagnostics would facilitate determining the root cause of an error or difference. Second, we plan to extend the evaluation of the effects of data compression on climate data in Baker et al. (2014) to ocean model data and will use the testing methodology presented here to evaluate the impact. The ability to determine whether

changes in the ocean state are statically significant or not due to data loss during compression is critical to the acceptance of compression as a tool to reduce data volumes for ocean simulation data.

## 7   Code availability

The CESM-ECT Python tools (pyCECT v2.0) can be obtained independently of CESM from NCAR's public git repository (https://github.com/NCAR/PyCECT/releases). The version of CESM used for our experiments, CESM 1.2.2, is available at

http://www.cesm.ucar.edu/models/cesm1.2. The CESM-ECT software tools are also included in the CESM public releases, with the POP-ECT addition available starting with the CESM 2.0 release series. CESM-POP simulation data is available from the corresponding author upon request.

*Acknowledgements.* This research used computing resources provided by the Climate Simulation Laboratory at NCAR's Computational and Information Systems Laboratory (CISL), sponsored by the National Science Foundation and other agencies. This work is supported in part

by a grant from the National Natural Science Foundation of China (41375102) and the National Grand Fundamental Research 973 Program of China (No. 2014CB347800).





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

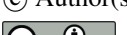



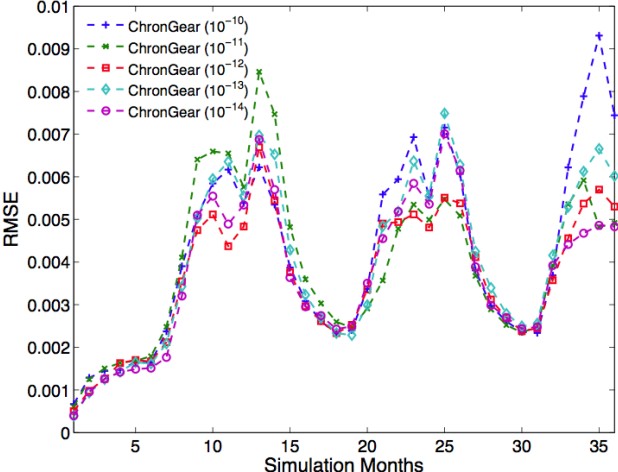

**Figure 1.** Monthly Root Mean Square Error (RMSE) of temperature for experiments with different barotropic solver convergence tolerances. Note that this is a subset of Fig. 12 in Hu et al. (2015).





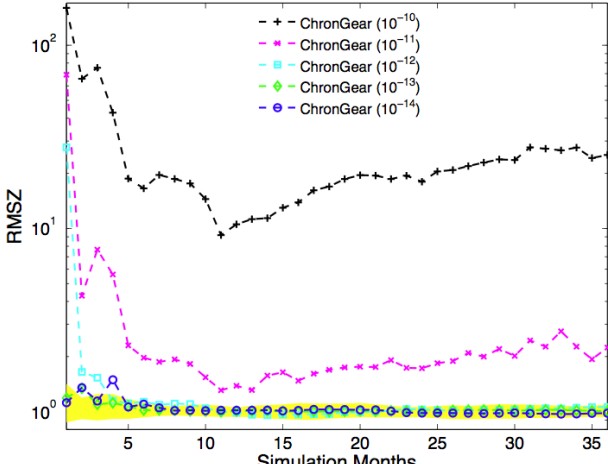

**Figure 2.** Monthly Root Mean Square Z-score (RMSZ) of temperature with respect to an ensemble (denoted by yellow) for experiments with barotropic different convergence tolerances. Note that this is subset of Fig. 13 in Hu et al. (2015).





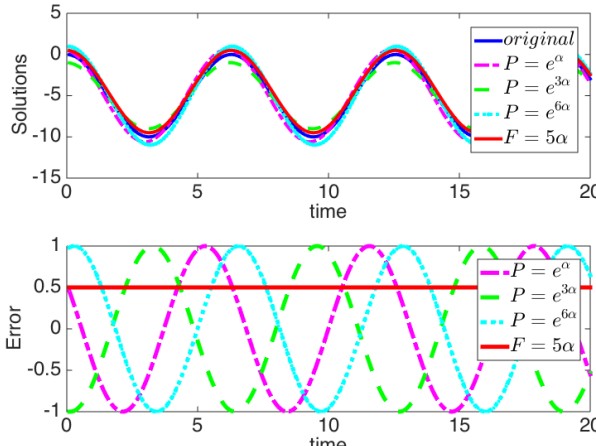

**Figure 3.** The top panel displays four analytic solutions to Eq. (2) with the indicated perturbations to the initial conditions (P) and perturbation to the forcing term (F). Note that all four perturbations have similar effect on the "original" (unperturbed) solution. The bottom panel plots the error between the four perturbations and the original solution. In this example, the error due to perturbing the forcing term (F) is constant and smaller in magnitude that the errors caused by the initial condition perturbations.





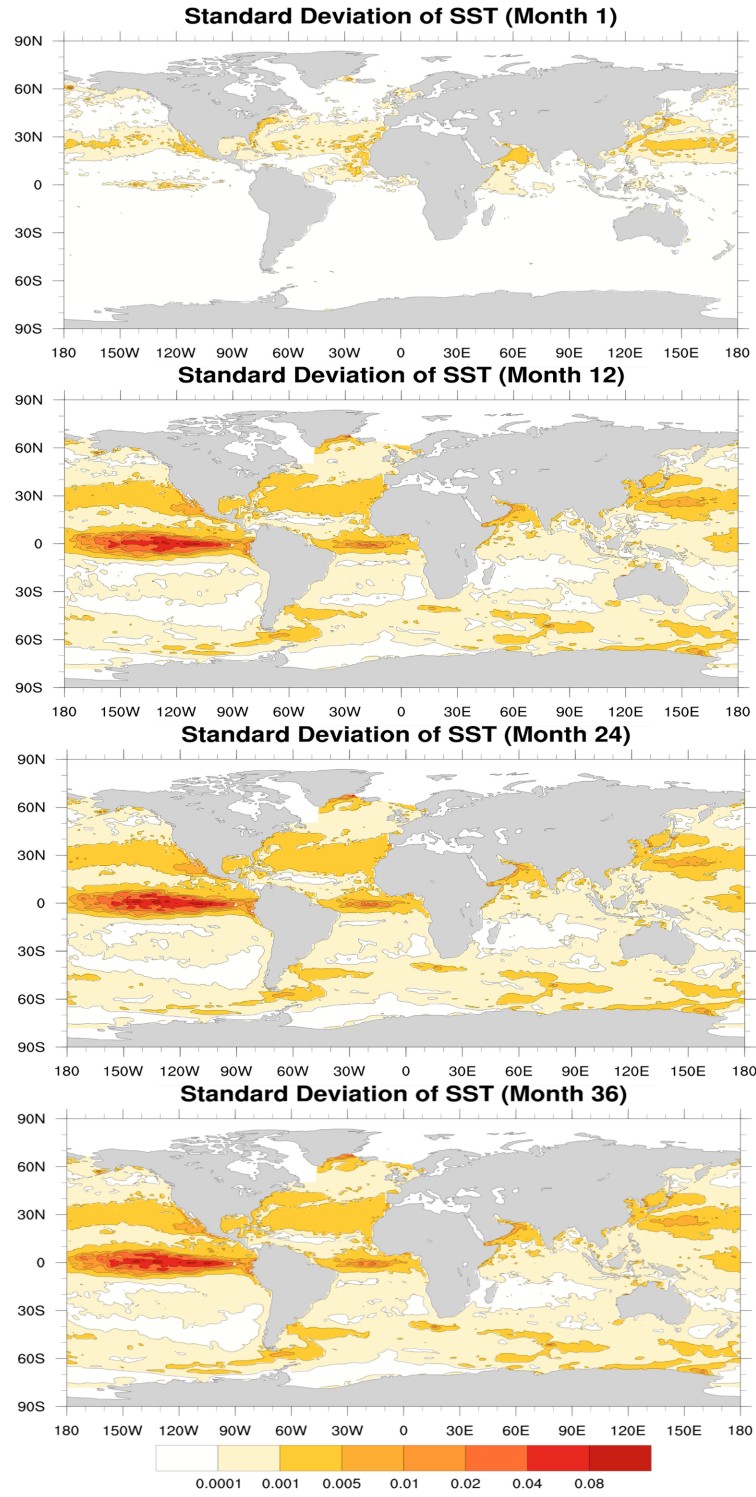

**Figure 4.** The ensemble distribution for the standard deviation of sea surface temperature (SST) at months 1, 12, 24, and 36.





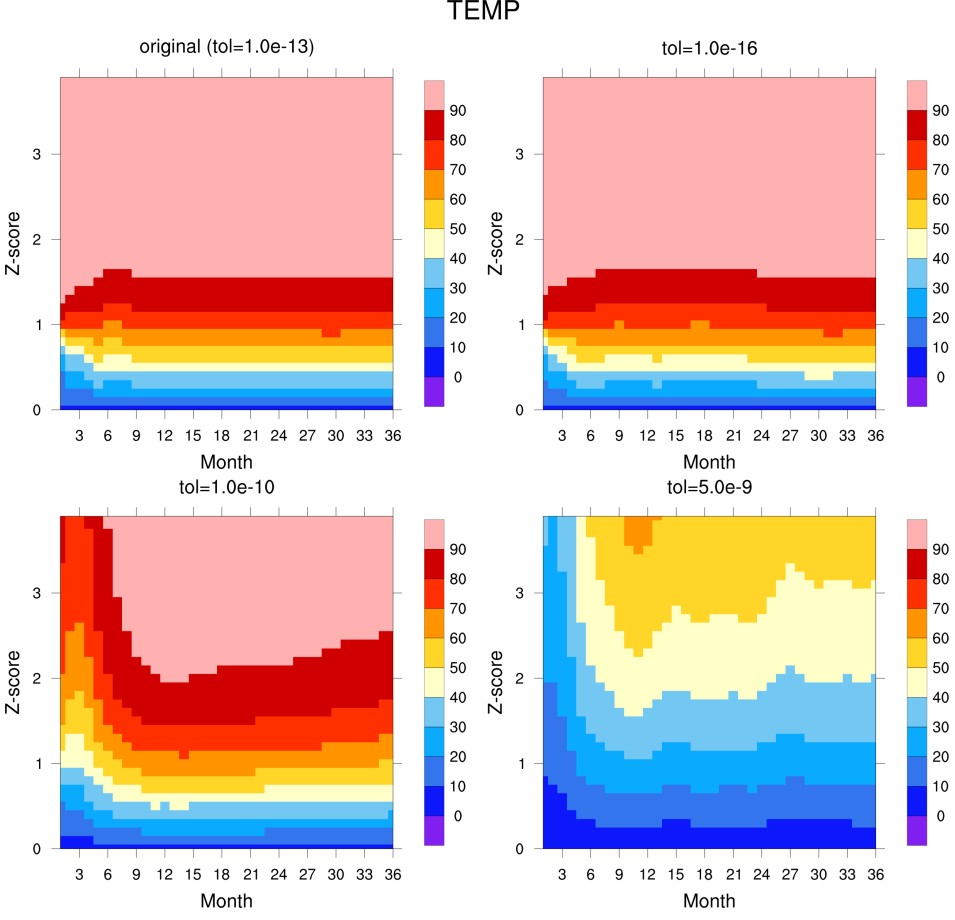

**Figure 5.** Response surfaces for Z-score of temperature (TEMP) over time (monthly) and Z-score tolerance. Each subplot represents a different barotropic solver convergence tolerance (labeled above). The color bar indicates the percentage of grid points with a Z-score below the Z-score tolerance (on the y-axis).

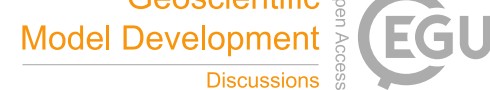



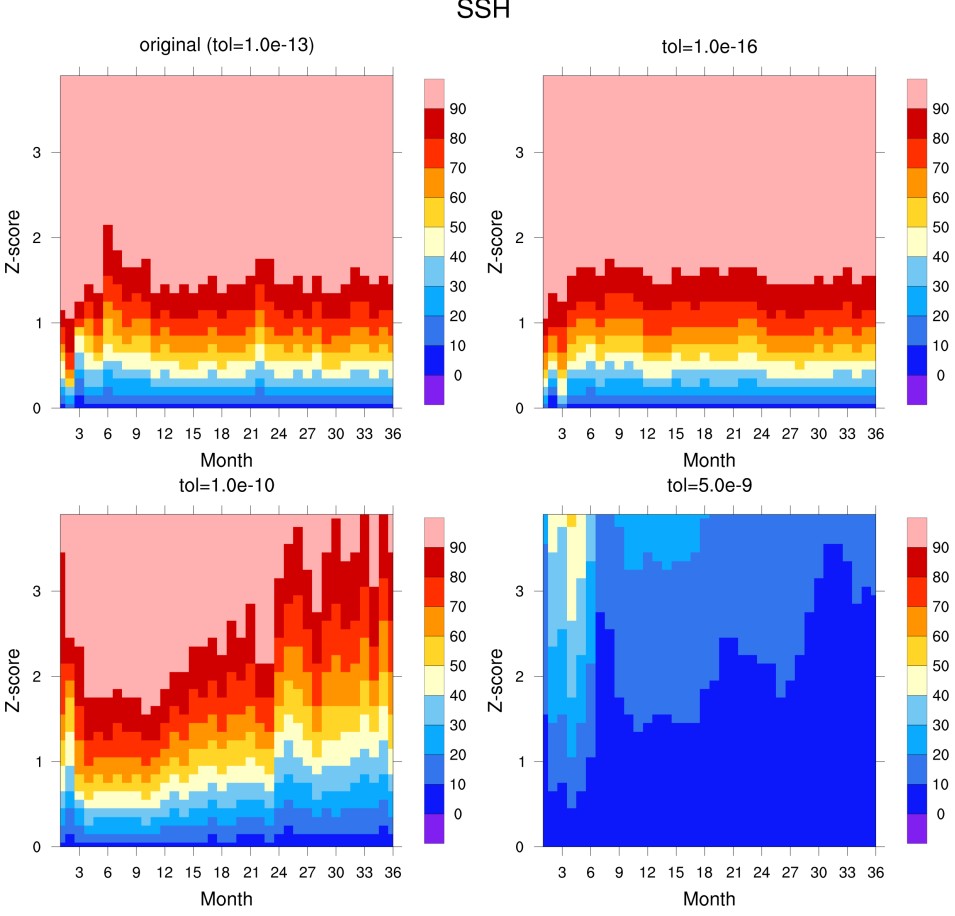

**Figure 6.** Response surface for Z-score of sea surface height (SSH) over time (monthly) and Z-score tolerance. Each subplot represents a different barotropic solver convergence tolerance (labeled above). The color bar indicates the percentage of grid points with a Z-score below the Z-score tolerance (on the y-axis).





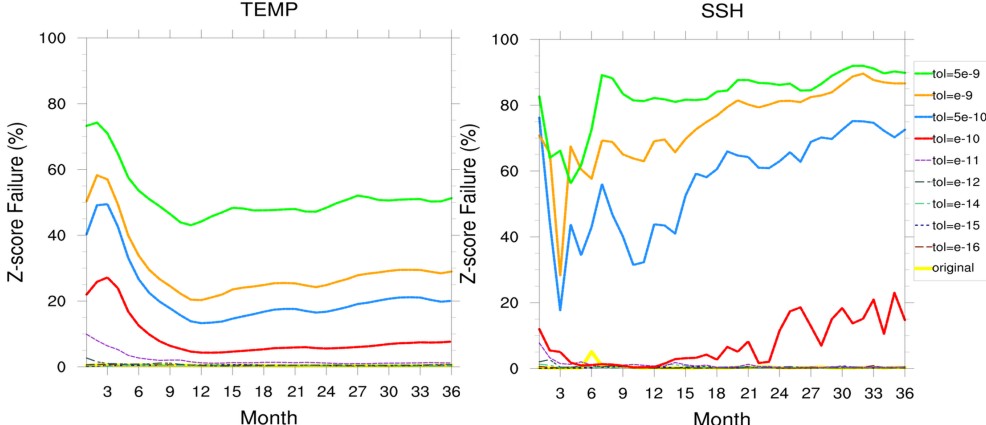

**Figure 7.** Percentage of grid points with Z-scores for temperature (TEMP) and sea surface height (SSH) that exceed the 3.0 tolerance for simulations with various barotropic solver convergence tolerances.

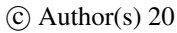



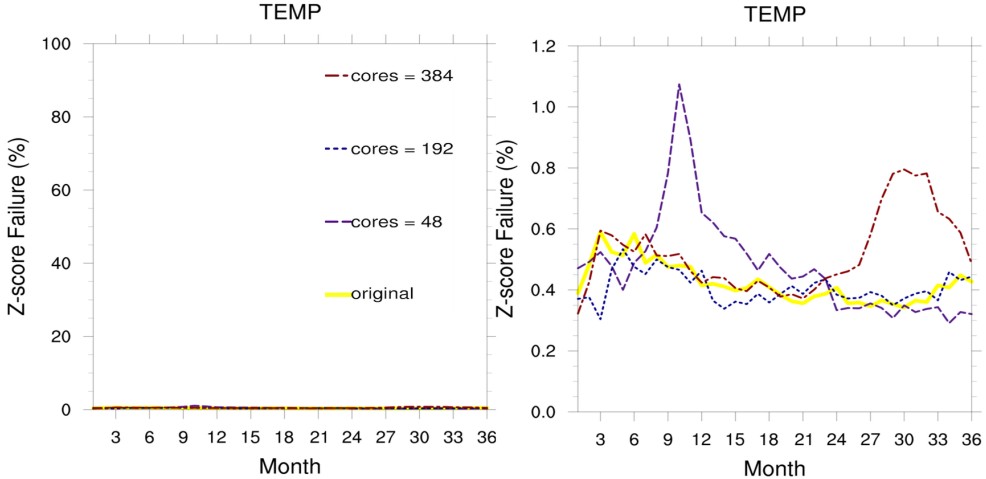

**Figure 8.** Percentage of grid points with Z-scores for temperature (TEMP) that exceed the 3.0 tolerance for simulations with various numbers of processor cores. (Note that the left and right subplots contain the same information, with different scales for the y-axis.)





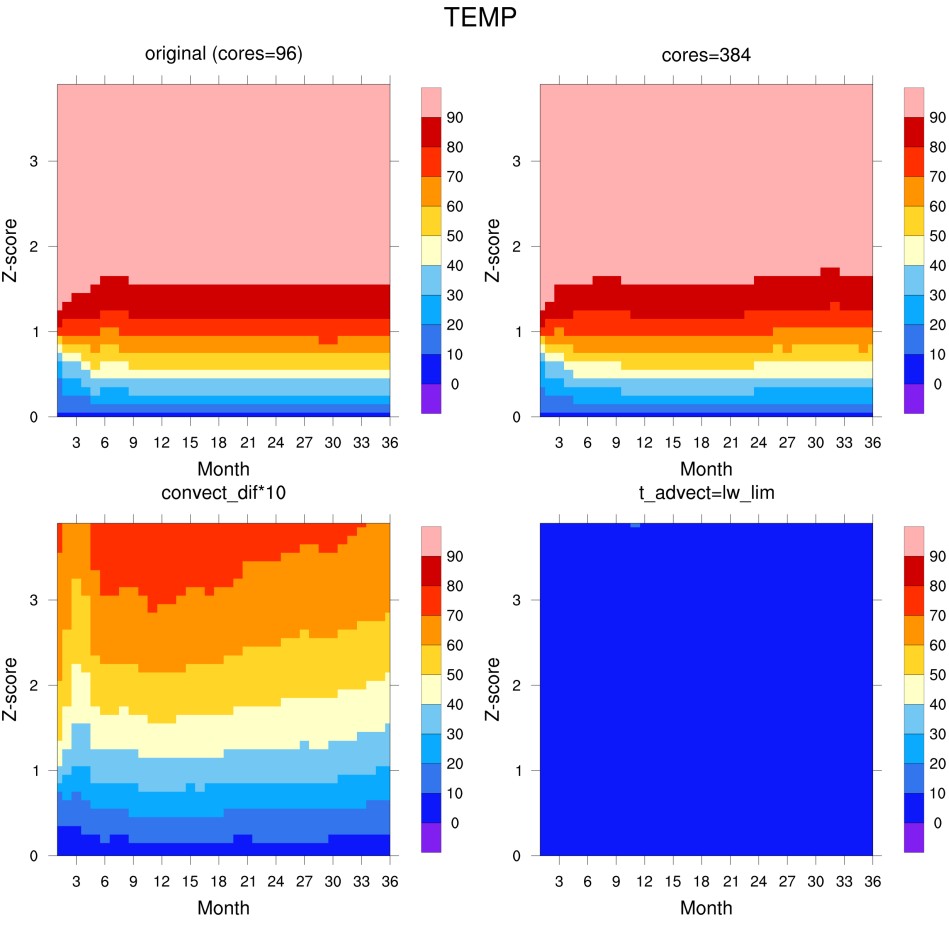

**Figure 9.** Response surfaces for Z-score of temperature (TEMP) over time (monthly) and Z-score threshold. The top two subplots represent two different processor core layouts. The bottom left has a tracer mixing coefficient for convective instability that is 10 times larger than the original, and the bottom right uses a different tracer advection scheme. The color bar indicates the percentage of grid points with a Z-score below the Z-score tolerance (on the y-axis).




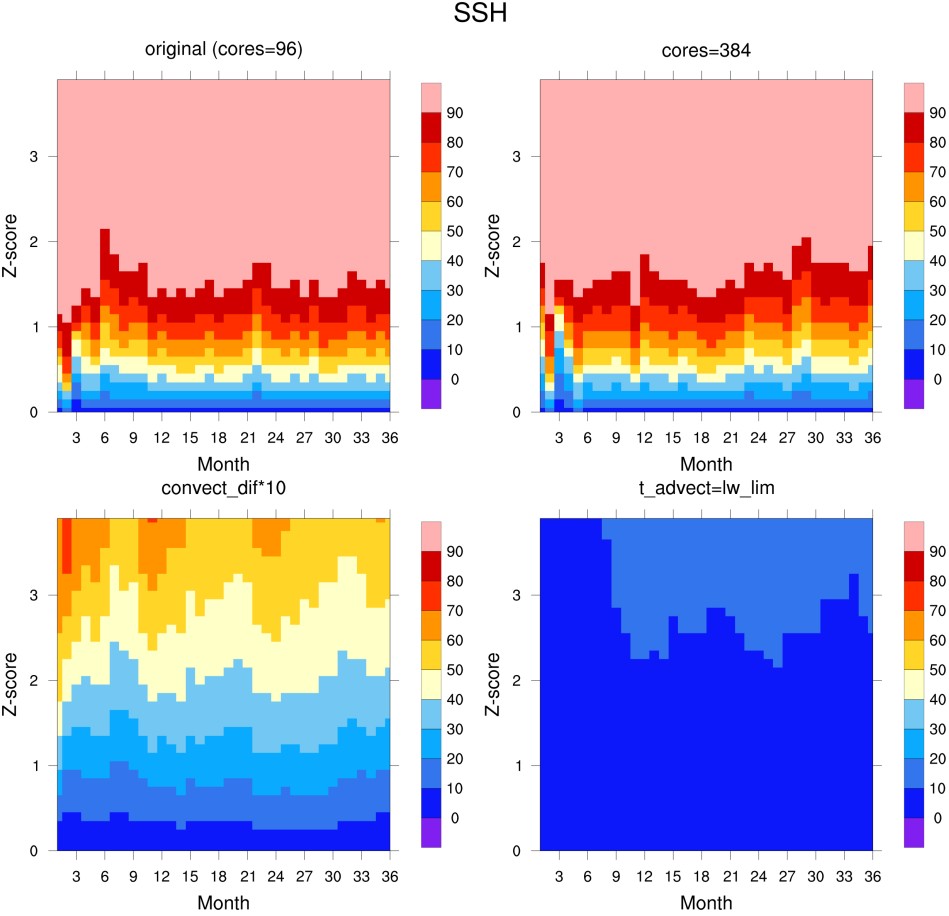

**Figure 10.** Response surfaces for Z-score of sea surface height (SSH) over time (monthly) and Z-score threshold. The top two subplots represent two different processor core layouts. The bottom left has a tracer mixing coefficient for convective instability 10 times larger than the original, and the bottom right uses a different tracer advection scheme. The color bar indicates the percentage of grid points with a Z-score below the Z-score tolerance (on the y-axis).





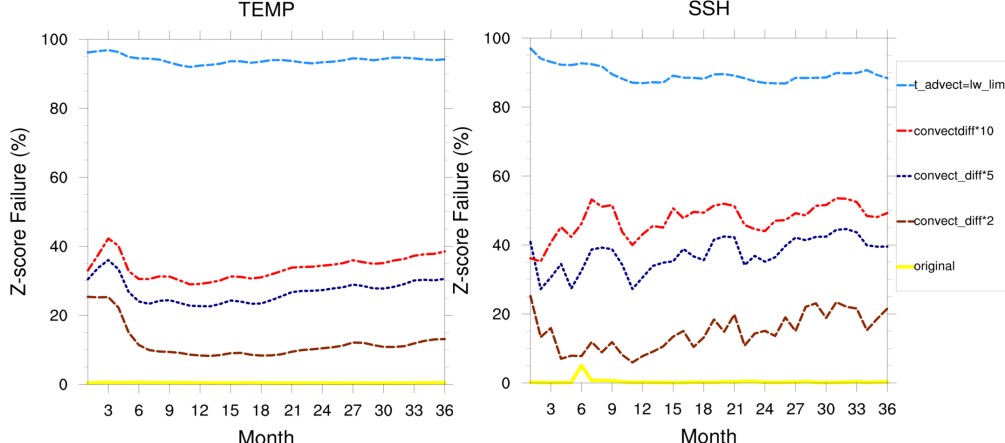

**Figure 11.** Percentage of grid points with Z-scores for temperature (TEMP) and sea surface height (SSH) that exceed the 3.0 threshold for simulations with an alternative tracer advection scheme (lw_lim) and several tracer mixing coefficients for convective instability.



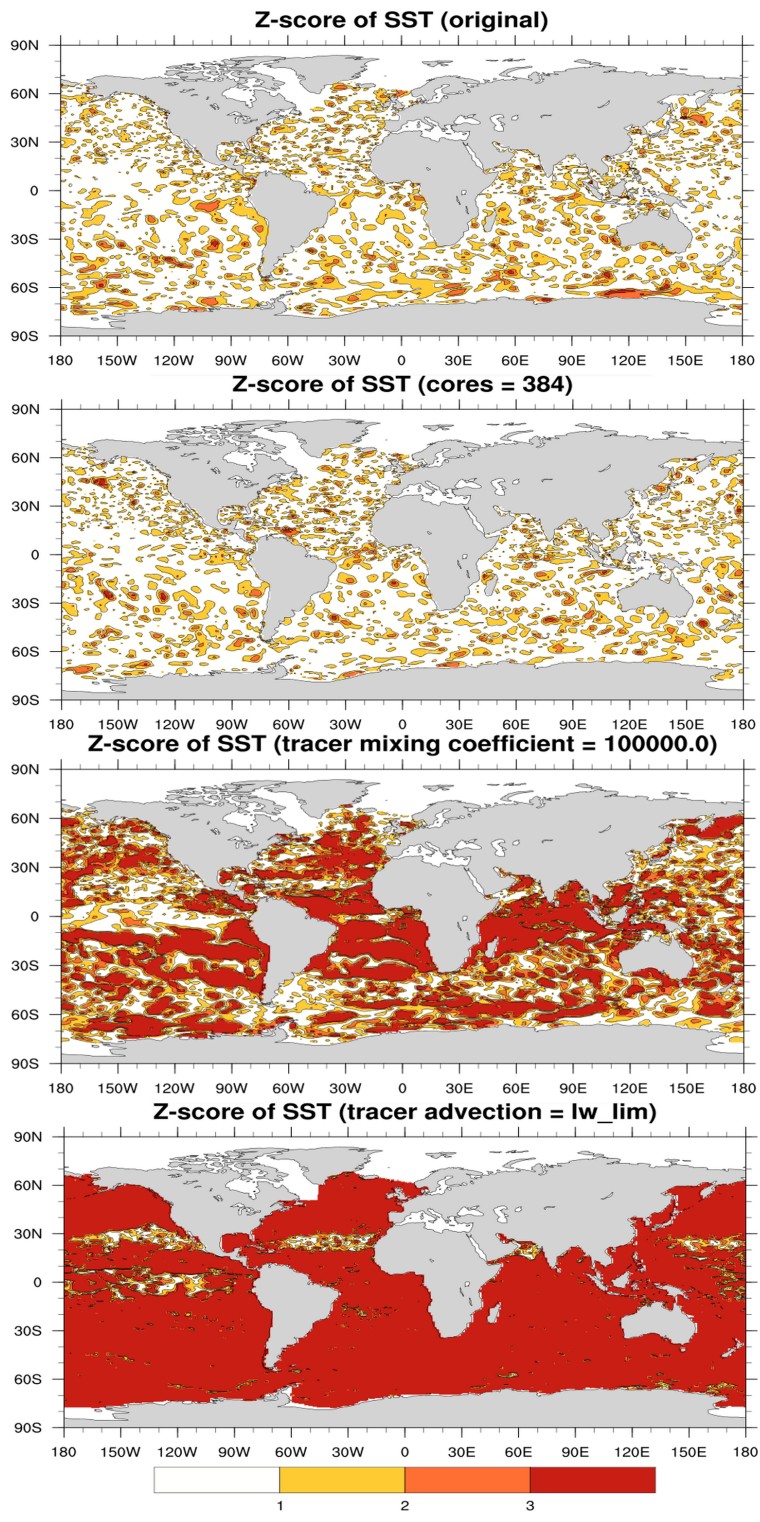

**Figure 12.** Z-score of sea surface temperature (SST) at month 12 for the original (default) case, a 384 processor core case, a case with a larger tracer mixing coefficient for convective instability, and a case with an alternate tracer advection scheme.




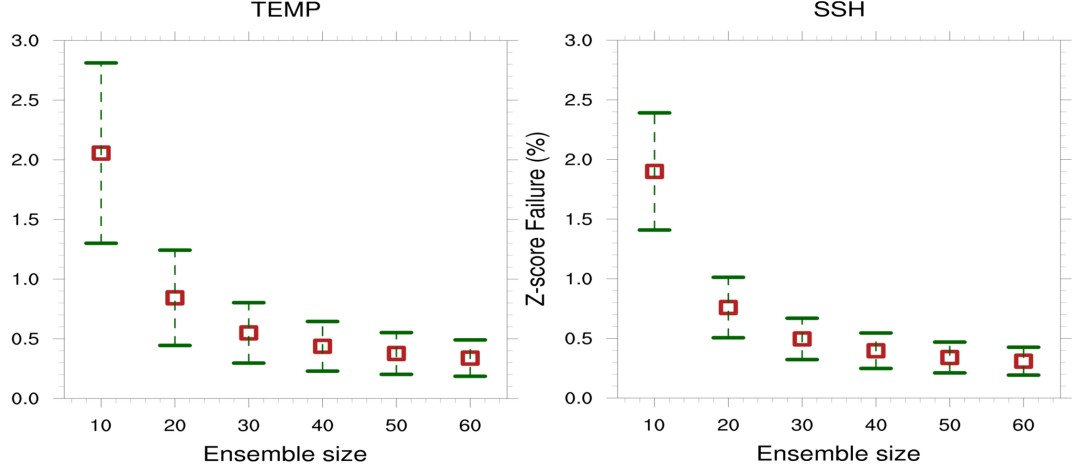

**Figure 13.** The distributions of experimental failure rates based on 1000 tests for variables temperature (TEMP) and sea surface height (SSH). For each ensemble size, the green bars indicate the maximum and minimum values obtained, and the red boxes indicate the mean.