# Peer review of "Evaluating Statistical Consistency in the Ocean Model Component of the Community Earth System Model (pyCECT v2.0)"

_Geoscientific Model Development, 2016_

## Referee Comment (RC1) · Anonymous Referee #1 · 22 Mar 2016

This paper tackles the important problem of developing statistical tests of "consistency" when a model is run to evaluate its climate. The fundamental problem is to develop tests on a chaotic system with sensitive dependence on initial conditions: the instantaneous state will obviously diverge under perturbation, but has it moved into a different basin of attraction? Current practice in most climate models is now brute force – run the model "long enough", a period determined by experience, custom, and institutional memory – and evaluate the results to see if the climate changed. This step is often subjective, and is in the eye (and metrics) of the beholder.

Baker et al (2015) attempted to solve this for the atmosphere. There an ensemble based on perturbing the initial conditions was used as the reference (151 1-year runs).

[Figure]

A PCA of 120 diagnostic fields is used (the 150-member ensemble is needed because N_ens > N_var is required for the PCA). That method showed reasonable success when applied to atmos-only runs of CAM, with some rate of false positives.

The ocean is considerably more difficult because of its different dynamical characteristics, and low-frequency modes of variability. This paper attempts to refashion the CAM-ECT method of Baker et al (2015) to CESM's ocean POP.

Unfortunately, this new paper is much less convincing than the old. There are several issues:

1. The setup is of an ocean-ice configuration driven by specified atmospheric BCs (what the authors call a "data model" p10L6). Those BCs are annually repeating, thus eliminating interesting modes of ocean variability, including basic coupled modes like ENSO. (An interesting and rather disturbing paper, Wittenberg 2014 in GRL, suggests that ENSO climate and other lower-frequency modes actually do not meet "climate consistency" as defined in this paper, even with the identical model!)

Furthermore it's subject to strong restoring of the salinity (actually its counterpart, freshwater) at the surface (p8L10). And at the resolution of the test, the model is rather dissipative (p7L35).

Thus the setup is a "low bar" compared to coupled models run for say, IPCC: dissipative and damped, with low-frequency modes filtered out. It's a necessary but not obviously sufficient condition of climate consistency of models as run in practice.

2. The test criteria ( $< 3\sigma$ over 90\% of the open ocean) is clearly a function the setup in (1). As is the ensemble size of 40 ... while it's well justified by the arguments of Sec 5 and Fig 13, it's not obvious how you'd extrapolate from the results of this paper to pick one for a different model, or even POP at a different resolution. All of the criteria of passing are very much dependent on the model and the setup, making the result rather weaker than one would hope. It might be strengthened if they could also

run a similar test with the hi-res POP configuration, and see how their criteria change. But perhaps that is too expensive, in which case one would question how this method would be used going forward.

3. The test cases include reproducing and strengthening the results of Hu et al for a different barotropic solver formulation; changing processor count; changing physics known to produce different climate. An additional test that might be interesting is to change the optimization level of the compiler, or to compare across different hardware as in Rosinski and Williamson. A false-positive there would be an interesting result.

4. These points above constitute very major revisions, in the opinion of this reviewer. Therefore I am not listing minor error or language improvements. However, I found Sec 2.2 very puzzling and it contains some fundamental errors. Eq 2 is not a solution of Eq 1, as elementary differentiation will show, nor does Eq 2 satisfy the boundary condition $X(0) = X\_0$. Eq 1 is also an ODE not a PDE (so the \del should be a d). And I think they have confused the steady-state solution of Caya et al (from where Eq 1 is drawn) with what they are trying to establish.

The point of this section seems to be that perturbing initial conditions and perturbing other model parameters can be treated as equivalent. That is true under some conditions, and many papers from the data assimilation literature (including ensemble approaches to DA like Kalman filter) will show you how and under what conditions the two are equivalent. The authors should cite that literature but in this reviewer's opinion, the whole of Sec 2.2 as written, and the associated Fig 3, will have to go.

I hope these comments are constructive, and a stronger paper will emerge from the review process, as the problem Baker et al are attempting to solve is both fundamental and in need of a solution.

---

## Referee Comment (RC2) · Anonymous Referee #2 · 26 Mar 2016

General comments:

Kudos for addressing an important and often overlooked challenge within geophysical model development - how to evaluate software and hardware changes being made to a chaotic system.

We need more work like this that is frank about shortcomings in model software quality.

Keep in mind that testing for bit for bit reproducibility across code changes is an extremely useful technique during model development. A large proportion of code changes should have no effect on model output. Since the test for BFB reproducibility can be done very easily it allows code changes that do not change results to be

merged quickly.

Specific comments:

In the introduction the exact circumstances in which pyCECT should be used are not made clear. I suggest adding a comment to indicate when pyCECT should _not_ be used. For example all code changes can be divided into commits that should change results and those that shouldn't. Using pyCECT to evaluate a change that shouldn't change results (but does) would be a mistake.

A good example of this is at 10: "or CESM-POP, even selecting a different number of cores on the same architecture results in non-BFB identical output". There are two reasons why this could happen 1) irreproducibility introduced by nondeterminism in MPI communications and 2) bugs in any of domain decomposition/ halo handling and possibly indexing errors. The correct way to handle this is to remove the 1) for example using "An order-invariant real-to-integer conversion sum" Hallberg and Adcroft 2014 and then fix the bugs in 2). So in fact I don't think changing domain decompositions is a situation where pyCECT should be used, this is just masking a problem.

In regard to this it may be worth noting that unexpected changes in model results across identical runs is an indicated of a software bug/problem (caused by, for example, invalid reads of uninitialized memory). The behavior of these bugs is unknown, they may have very little impact most of the time (otherwise they would have been found) but occasionally significantly change results. This can all depend, for example, on the contents of uninitialized memory.

Similar to the above points. pyCECT should not be used as a substitute for proper (unit) testing. You give the example of introducing a new solver to improve performance. It would be tempting to skip the unit testing and use pyCECT with the new solver and declare that it's bug-free because the results are not statistically different.

The value of section 4.2 is questionable given the above points. My conclusion from

this would be the opposite of yours. i.e. it shows that the simulation differences are noticeable when there should be none.

Technical notes:

The plot layout should be improved. For example, in figure 9, can you include a short description or label for the color bar in the plot itself. Also, instead of labels like convect_diff*10, t_advect=tw_lim can you use descriptive names.

Thoughts (but no changes required):

It would be nice to have some mention of the effects of model resolution here. Increasingly climate models are using ocean components higher than 1deg. Wouldn't it make sense to perform pyCECT analyses on runs which are as low as possible resolution (for performance reasons)? if so what are the constraints on using pyCECT in this way?

---

## Author Comment (AC1) · 17 Apr 2016

We very much appreciate the thorough review and suggestions for improvement. We address the four main issues below.

(1) Model configuration concerns

We agree that POP-ECT cannot test for "climate consistency" as specified by the reviewer for low-frequency mode ocean events. Indeed, many works emphasize that the ENSO and low-frequency modes do not meet the climate consistency as defined here. A well-known example for our CESM framework is the CESM Large Ensemble Project (LENS) which perturbs SST at O(-12) round-off level (Kay et al., 2015 - see

below) and takes advantage of this inconsistency driven by the small initial differences to establish the large ensemble bases. However, the purpose of POP-ECT is to test for consistency with an established ensemble, and its main application is to identify potential problems associated with software or hardware configurations, not primarily scientific exploration.

We agree that our design minimizes the natural variability introduced by the surface boundary conditions and other potential forcing by using the climatological data-driven forcing. ENSO or low-frequency variability simulations will fail the POP-ECT if coupled simulations are conducted because of the chaotic behaviors in the atmosphere model. We also agree that the current setup is the "minimal" requirement to detect if the new run is consistent with the ensemble or not because the oceanic model is much more dissipative than the atmospheric model, which is why the approach of Baker et al. (2015) cannot be directly applied to the ocean component (see our discussion in section 1). In our current setup, a failing result from the POP-ECT can inform the user that a code or environment (hardware/software) change made by the user may be problematic. We note that the purpose of both Baker et al. (2015) and this manuscript is not to evaluate the climate consistency in the coupled climate production simulation. We are instead trying to identify potential errors induced during the software development lifecycle, such as porting to a new machine architecture, optimizing the code, changing compilers, or modifications to the machine hardware or software stack, etc. We have clarified our intent in the revision by adding text to the abstract, introduction, and conclusion.

Kay, J. E., Deser, C., Phillips, A., Mai, A., Hannay, C., Strand, G., Arblaster, J., Bates, S., Danabasoglu, G., Edwards, J., Holland, M. Kushner, P., Lamarque, J.-F., Lawrence, D., Lindsay, K., Middleton, A., Munoz, E., Neale, R., Oleson, K., Polvani, L., and M. Vertenstein (2015), The Community Earth System Model (CESM) Larg Ensemble Project: A Community Resource for Studying Climate Change in the Presence of Internal Climate Variability, Bulletin of the American Meteorological Society, doi: 10.1175/BAMS-D-13-00255.1, 96, 1333-1349. http://journals.ametsoc.org/doi/abs/10.1175/BAMS-D-13-00255.1

(2) Test criteria

We completely agree that the test criteria is a function of the setup, and it is not obvious that the same criteria would be appropriate for tenth degree resolution, for example. This same comment applies to our previous CAM-ECT work. That said, because the tool is specifically intended to verify a new CESM hardware setup environment or a minor code modification intended to result in a consistent climate (e.g. reordering operations in a stable and mathematically equivalent way for optimization purposes), in most cases a single test configuration will be sufficient. One cannot hope to test every possible configuration and resolution, but an error in the software or hardware will likely manifest itself regardless of the configuration. If, however, one made a code change that only affected high resolutions, then POP-ECT with a low resolution would not catch such an error. We would argue, though, that such testing should be done in the context of software unit testing, however - not via this ensemble consistency evaluation. Upon re-reading the manuscript, we realized that we did not make the intent of the tool clear enough and have updated the text accordingly.

(3) Test cases

We agree that additional tests that evaluate a change in the hardware or software stack are of interest. In the revision, we have added a new section that includes results from the following modifications:

-changing the compiler on the same machine (e.g., GNU or PGI instead of Intel) - changing the machine (e.g. NERSC's Edison machine) -changing the compiler version or optimization flag

(4) Section 2.2

We agree that there are errors in this section. We have decided to remove most of this

section, as it is tangential to our focus. We left only the text describing work done in Hu et al. 2015 for background.

———————————————————

---

## Author Comment (AC2) · 17 Apr 2016

Thank you for your thorough review and suggestions for improvement. We address all comments below.

**\*\*General comments\*\***

Thank you for the "kudos". Indeed many of the CESM code changes retain BFB reproducibility and are easily verified. This new tool is currently mostly used for verifying ports to new machines.

**\*\*Specific comments\*\***

(1) circumstances in which pyCECT should be used

We agree and updated the Introduction in the revision to clarify when pyCECT should be used (and when it should not).

(2) non-BFB identical output due to nondeterminism in MPI communications

We agree that ideally this situation is avoided via "customized" deterministic MPI routines. Note that such functionality currently exists in the CESM atmosphere component (CAM). However, because this functionality does not currently exist in POP (nor is it realistically likely to be added in the near future), verifying that a decomposition change in POP does not result in an inconsistent result is important for POP, in our opinion.

(3) unexpected changes in model results across identical runs

We agree that changes in model results across identical runs are often symptomatic of a bug or problem. However, this is not necessarily the case. For example, on Blue Waters (at NCSA), CESM does not produce reproducible results when the FMA (fused multiply-add) capability is used. In general, newer heterogeneous architectures will make reproducible results difficult to obtain, particularly if one takes advantage of the optimizations provided (e.g., FMA).

(3) unit testing

We agree that pyCECT should *not* be used as a substitute for unit testing. Unit testing is important and an integral part of software quality assurance. We made a comment to this effect in the revision.

(4) section 4.2

We retained section 4.2 for the reason stated in (2) above. Note that we also added additional experiments (in response to another reviewer) that test changing the machine and compiler.

**Technical notes**

(1) plot layouts

We improved the labels and added the suggested color bar descriptions.

**Thoughts***

(1) model resolution

Computational cost is certainly a consideration when considering high-resolution models, and we plan to address this in future work.

---

## Author Response (AR2)

* * *
Response to second review by reviewer RC1
* * *
We appreciate the reviewer's helpful suggestions for improvement.  We address the comments below.

Major modifications:

1) Re: "Some of the responses to RC1 in AC1 need to be included in the text."

To address this concern we added an additional section, entitled "Additional discussion: scope and limitations" that is now Section 6.

Re: "Please include the Deser citation as well."

By "Deser" citation, we assume that the reviewer meant the Kay, Deser, et al. citation that we listed in our initial AC1 response.  This citation (Kay, 2015) has been added.

2) Re: "The answer (2) in AC1 is also adequate but the article should state clearly that findings are resolution-dependent."

We now reiterate this point in the newly introduced section (see answer above), entitled "Additional discussion: scope and limitations".

We also removed the reference to the CAM-ECT ensemble size on  P5L11.  However, note that for CAM-ECT, $N\_ens > N\_var$ only establishes a lower bound.

3) Re: "Section 4.5 is a welcome addition. But is it clear that POP-ECT is an advance over POP-RMSE (which is considerably cheaper)? "

We are pleased that the addition of section 4.5 was well-received.  We believe that the additional experiments significantly improved the results section.

We note that the RMSE test has already been shown to be ineffectual for distinguishing convergence tolerance differences in Yong 2015, and in this

paper we discuss that result in section 2.1 (and also show a subset of the Yong 2015 results in Fig. 1).  We do not feel that additional insight will be gained by providing such plots for the all the other test example failures.  If the reviewer is instead interested in 5-day output (as in POP-RMSE), we would need to re-do all of our experimental runs as only monthly output was saved.  Note that results from POP-RMSE are subjective: there is not a simple pass/fail criteria. Given the inconclusive nature of this test, it is not clear that the resources required to include such results would contribute significantly to the quality of the manuscript.

Minor comments:

All have been corrected in the revision.
* * *
Manuscript Changes
* * *
Section 2.4:
Removed the reference to the CAM-ECT ensemble size.

Section 6:
Added a new section to clarify scope and limitations. Added a reference to Kay 2015.

Minor:
Corrected mispelling of "consistent" and redid sentence P5L20.

[revised manuscript text omitted]

Velocity in grid-x direction difference (GNU vs. Intel)

[Figure]

Velocity in grid-x direction difference (PGI vs. Intel)

[Figure]

**Figure 12.** Differences in the top-level zonal velocity (UVEL) in cm/s at month 12. The top plot shows the differences between the GNU 14.8 and default Intel simulation ouput. The lower plot shows the differences between PGI 13.9 and the default Intel 13.1.2 simulation outputs. Note that the min, max, and mean data values for UVEL for the default Intel 13.1.2 simulations at month 12 (top-level) are -80.9, 77.0, and -0.8 cm/s, respectively. For comparison, the min, max, and mean data values for UVEL are -87.9, 69.8 and -0.9 cm/s for the PGI 13.9 simulation data.

[Figure]

**Figure 13.** The distributions of experimental failure rates based on 1000 tests for variables temperature (TEMP) and sea surface height (SSH). For each ensemble size, the green bars indicate the maximum and minimum values obtained, and the red boxes indicate the mean.